# Robust Fastener Detection Based on Force and Vision Algorithms in Robotic (Un)Screwing Applications

**DOI:** 10.3390/s23094527

**Published:** 2023-05-06

**Authors:** Paul Espinosa Peralta, Manuel Ferre, Miguel Ángel Sánchez-Urán

**Affiliations:** Centre for Automation and Robotics (CAR) UPM-CSIC, Universidad Politécnica de Madrid, 28006 Madrid, Spain; paul.espinosa.peralta@alumnos.upm.es (P.E.P.); miguelangel.sanchezuran@upm.es (M.Á.S.-U.)

**Keywords:** fastener detection, unfastening process, robot vision, socket wrench, segmentation neural networks

## Abstract

This article addresses how to tackle one of the most demanding tasks in manufacturing and industrial maintenance sectors: using robots with a novel and robust solution to detect the fastener and its rotation in (un)screwing tasks over parallel surfaces with respect to the tool. To this end, the vision system is based on an industrial camera with a dynamic exposure time, a tunable liquid crystal lens (TLCL), and active near-infrared reflectance (NIR) illumination. Its camera parameters, combined with a fixed value of working distance (WD) and variable or constant field of view (FOV), make it possible to work with a variety of fastener sizes under several lighting conditions. This development also uses a collaborative robot with an embedded force sensor to verify the success of the fastener localization in a real test. Robust algorithms based on segmentation neural networks (SNN) and vision were developed to find the center and rotation of the hexagon fastener in a flawless condition and worn, scratched, and rusty conditions. SNNs were tested using a graphics processing unit (GPU), central processing unit (CPU), and edge devices, such as Jetson Javier Nx (JJNX), Intel Neural Compute Stick 2 (INCS2), and M.2 Accelerator with Dual Edge TPU (DETPU), with optimization parameters, such as the unsigned integer (UINT) and float (FP), to understand their performance. A virtual program logic controller (PLC) was mounted on a personal computer (PC) as the main control to process the images and save the data. Moreover, a mathematical analysis based on the international standard organization (ISO) and patents of the manual socket wrench was performed to determine the maximum error allowed. In addition, the work was substantiated using exhaustive evaluation tests, validating the tolerance errors, robotic forces for successfully completed tasks, and algorithms implemented. As a result of this work, the translation tolerances increase with higher sizes of fasteners from 0.75 for M6 to 2.50 for M24; however, the rotation decreases with the size from 5.5° for M6 to 3.5° for M24. The proposed methodology is a robust solution to tackle outliers contours and fake vertices produced by distorted masks present in non-constant illumination; it can reach an average accuracy to detect the vertices of 99.86% and the center of 100%, also, the time consumed by the SNN and the proposed algorithms is 73.91 ms on an Intel Core I9 CPU. This work is an interesting contribution to industrial robotics and improves current applications.

## 1. Introduction

One of the most widely demanded tasks in the industry is screwing and unscrewing. However, this precision task continues to be challenging for robots, especially when the environment changes and the requirements increase. Therefore, to perform a task with high reliability, it is necessary to use additional sensors such as vision and force. These sensors make it possible to automatize robotic tasks. However, more advanced algorithms are required to adapt dynamically to the environment. Some examples include screwing/unscrewing in small workspaces, where it is not possible to have a dispenser, or when a different-sized screw or high screw torque is required. Another drawback of this task is when the environment is unstructured and the robot needs to insert the fastener or nut, which can be complex from the viewpoint of robotics and when the probability of failure is high [1]. Another problem is working in an outside environment with different illumination conditions and difficult access for humans [2,3]. A further complicated task in unstructured illumination is the localization and maintenance of fasteners in railways [4,5,6]. This is, therefore, an active research area where various methods are in continuous development, especially algorithms or control strategies based on force and vision. Nonetheless, each strategy has its advantages and disadvantages, as summarized in Table 1. In [7,8], the authors used a force sensor and an active compliance strategy to engage the tool on the screw, but the search time is inversely proportional to the fastener’s size, which, in some cases, can be very time-consuming. The authors of [9] proposed using robot fingers to position, grasp, and unfasten unknown random-sized hexagonal nuts. In other works, such as [10,11], industrial cameras with external illumination were used. The lights help maintain consistent image contrast and set the camera focus. The drawback, however, is that they only allow for imaging at a defined distance, reducing the detection for some fastener sizes. However, others studies have used traditional vision and force algorithms to increase robustness [12,13,14], although these are susceptible to errors due to illumination variance, and vision algorithms may fail. The authors of [15] proposed a computer vision framework using a combination of neural networks for detecting screws and recommending related tools for disassembly. Reference [16] proposed unscrewing strategies that include conducting a second search and a second unfastening trial, as well as involving collaboration with a human operator.

The main goal of this work is to develop a robust, automatized system for screwing in several industrial applications under different environmental conditions. Most of the current systems just developed are specifically designed for predefined and controlled illuminated conditions. This study describes a more general solution that has been tested in outdoor and indoor applications and also with a set of fasteners from M6 to M24. The main components of the proposed solution are a robot that carries an industrial camera with a robust vision algorithm to detect several fasteners and a force sensor for monitoring the interaction force. This makes it possible to work in semi-unstructured environments where the position of the target is not precisely defined. The vision subsystem can be adapted to different fastener sizes by fixing the field of view to the workspace size. The combined use of vision and force sensors has demonstrated the excellent performance of the screwing system.

The vision subsystem is based on an industrial smart camera with a dynamic focus and active NIR illumination. The integration of both components has allowed working with several sizes of fasteners under harsh lighting conditions. Moreover, a collaborative robot with an embedded force sensor and a customized tool is used. For this application, robust algorithms based on SNN and vision were developed to find the center and rotation of the fastener. A comparative table of accuracy and FPS with some common SNNs is presented to describe the best model and device to use. In addition, a mathematical analysis of the manual socket wrench was performed to determine the maximum error allowed. These errors were experimentally verified when the tool fit perpendicularly to the fastener. The force of the robot was measured for each zone, and it defines the task status. In addition, the algorithms proposed were exhaustively tested under random and controlled illumination conditions in a synthetic dataset (1,000,000 images) with several fastener sizes (M6, M8, M10, M16, M24). Finally, a real application using an M10 fastener test was performed to verify the effectiveness of this work.

The structure of the article is as follows: Section 2 explains the related work. Section 3 describes the scenario and experimental setup. Section 4 details the process to obtain the real and theoretical rotation and translation tolerances using a manual socket wrench and different-sized fasteners. Furthermore, the limits of forces executed by the robot are considered. Section 5 presents the performance of SNN. Section 6 describes the methodology, test, and results of the proposed algorithm. Finally, Section 7 presents the conclusions.

## 2. Related Work

### 2.1. Robots in Screwing/Unscrewing Tasks

Several types of robots and tools are used in industry to perform screwing tasks. Some applications use a commercial tool to manually change the socket wrench. For example, in [7], a collaborative Kuka robot and the Renault nutrunner are used to disassemble the turbocharger. Reference [10] uses an industrial FANUC robot with the Desoutter nutrunner to assemble parts in the automotive industry. In other cases, a specific tool is designed and can allow for the automatic interchange of the socket wrench, such as in [13], which uses a KUKA LWR with a mechanical system for automatic bit change at the battery disassembly station. In [12], the authors designed a compact tool to unscrew nuts in overhead lines. A standard manual or impact socket wrench is commonly used in these tasks. Not all sockets are similar, however, and there are many types on the market. The choice of a specific one will depend on many factors, which are detailed in the following subsection.

### 2.2. Socket Wrench

Determining the tolerance error of the tool designed for screwing tasks is of great importance. Doing so determines the robot and camera that may be used to perform the task. One of the main components of the tool is a socket wrench; its manufacturing is determined by several norms based on ISO and patents, according to which three factors were determined to use closed-socket wrenches in robotic applications. The first is the number of polygonal vertices, either six or twelve. Six-point socket positions tighten fasteners every 60 degrees with high torque because they have a larger contact area with the fastener, whereas twelve-point sockets position at 30 degrees easier and faster but withstand less torque because their walls are thinner. The second norm is the maximum and minimum tolerances of sockets’ wenches and fasteners. In sockets, the tolerances to consider are hexagonal head size [17], square drive [18], rounded shape in the vertices (depending on the manufacturer’s patent), and chamfer angle [19], whereas in fasteners, hexagonal head size and chamfer angle [20] are considered. The third factor is six-or twelve-point sockets, which have groove angles to create complementary surfaces that act as driving surfaces to avoid contact with the corners and reduce the internal stresses exerted on the socket during driving. In some patents, the angles are constant values [21,22], and in others, the angles are based on the tolerances of the bolt and fastener [23,24].

### 2.3. Vision System

The vision system has many parameters to be considered, such as the following.

The WD to acquire the images with a specific FOV: Cameras with fixed focus only work with a specific WD, while cameras with autofocus can acquire images from different heights. Consequently, the FOV and pixel size differ. FOV can be calculated using the pinhole camera model [25]; this model describes the geometry of the optical devices, and the FOV is obtained using triangle similarity.Autofocus: Cameras with a variable FOV have an electrically adaptable liquid crystal lens (TLCL); this technology changes the electrical voltage to focus an image [26,27,28]. Furthermore, to obtain the correct focus value, there are many algorithms [29], with the most widely used in cell phones and industrial applications being phase detection autofocus systems (PDAF) and contrast detection autofocus systems (CDAF) [30,31].Pixel size: This is defined by the ratio of the FOV and camera resolution. However, this value is not accurate because the complete model of a camera has intrinsic and extrinsic parameters and distortions to consider. It is necessary to calibrate the camera [32] to find the exact value in a given WD.Time exposure: A robust vision system has to work with various illumination levels. Therefore, the use of active NIR illumination (wavelengths of [750 : 1000] nanometers) [33,34] and time exposure dynamic algorithms [35,36,37] enhances robustness.

### 2.4. Fastener Detection

Some works use classic vision algorithms to find a fastener when environmental illumination is controlled or constant [10,12,14,35], meanwhile in other cases, neural networks are used for classification [38], detection [39], and object segmentation [11,40]. State-of-the-art segmentation neural networks [41,42,43,44] are highly precise in finding objects under certain adverse illumination conditions. Furthermore, depending on the number of layers or depth of the model, as well as the type of the variable used to store the network weights (UINT8, INT16, FLOAT32), the inference will be more or less time-consuming.

Additionally, the trained model could be improved using image augmentation, which helps to mitigate the effects of the approximate values of autofocus and time exposure. The result of segmentation is a mask; in some models [42], it may have edges with ripples, whereas in other cases, could exist different images not considered in the trained dataset, such as fasteners with worn, scratched, and rusty heads. Thus, inferred masks and their edges may not be well defined, so additional vision algorithms [45,46,47] are needed to improve robustness and accuracy.

## 3. Scenario and Experimental Set-Up

### 3.1. Testing Set-Up

The test bench shown in Figure 1 consists of several subsystems, each of which performs a specific task:The Collaborative Robot, which performs specific trajectories and corrects the offset given by the vision system. The robot chosen is Universal Robots UR5e. With a maximum payload of 5 kg and ±0.03 mm ISO 9283 [48] pose repeatability. It is connected by the Profinet protocol with the PLC.The tool, it was designed to connect in the robot’s wrist 3, and contains a socket wrench to screw and unscrew the fasteners.Industrial smart camera Omrom F430-F000L12M-SWV [49] with CMOS monochrome sensor, 1.2 megapixels (1280 × 960), 16 mm focal lens, autofocus technology to enable work from 75 to 1500 mm, and NIR lead and filter (850 nm).It is connected via TCP/IP to the PC and can capture and download images.Aluminum plates, the first one with M6, M8, M10, M16, and M24 fasteners, and the second with M8 and M10 fasteners.PC Intel Core I9 10-Gen-900H [50] with a virtual PCL running in TwinCat-Beckhoff software, performing the master control, managing the communication of all components and serving as a data logger. The vision algorithms are executed in Python-TensorFlow, also, the ADS-Twincat is used to transfer data between PLC and Python script.Embedded devices to accelerate the neural networks, the following devices were tested: JJNX [51], INCS2 [52], and DETPU [53].Servomotor Qrob70 [54], and an electric linear actuator to introduce uncertain positions in rz, rx in a real test. The servomotor communicates through EtherCAT and the linear actuator by a serial-USB with the PLC. The uncertainties in rx are read with an IMU using a serial-USB.

### 3.2. Experiments Conducted

Two test benches have been developed for the experimental works related to this study. The first test bench was used for the experimental calculations of the translation and rotation tolerance. The second test bench focused on the computer vision algorithms for indoor and outdoor environments; more details about these test benches are as follows.

Test bench 1–Obtaining the experimental rotation, translation tolerances, and forces exerted when a robot inserts a socket into different fastener sizes.–Obtaining several photos according to the camera settings as a function of fastener sizes with a variable and constant fastener FOV, as well as manually obtaining the center and vertex as ground-truth values. Additionally, an analysis of the performance of the algorithm is conducted to simulate environmental conditions to obtain the proposed vertices and center.–Comparing the performance of the proposed algorithm to calculate the rotation with an approximation contour algorithm [55] using a synthetic dataset.Test bench 2–Testing the proposed methodology in outdoor environments under different light conditions and uncertain positions in the x, y, rx, and rz axes, using a specific fastener size.

### 3.3. Contributions

By means of the test benches, proposed algorithms, and experiments conducted, some significant contributions have been provided.

Mathematical analysis was performed to obtain translation and rotation errors between several sizes of manual socket wrenches and fasteners, according to the ISO tolerances and manufactured patents.Experimental translation, rotation tolerances, and force values were obtained in an unscrewing process using a collaborative robot with several fasteners and socket sizes.A comparative inference performance between the most common segmentation neural networks using GPU, CPU, and Edge devices (JJNX, INCS2,DETPU) in some formats (FLOAT32, FLOAT16, UINT8).A robust and novel vision algorithm was created to find the center and rotation of the hexagon fastener in flawless conditions, as well as worn, scratched, and rusty conditions in outdoor environments with varying lighting conditions.An exhaustive evaluation test was carried out to validate the algorithms implemented under simulated illumination and real conditions.

## 4. Tolerances in the Manipulation Tool

The main objective of the algorithms proposed in this work is to find the center and the rotation of the fastener with high precision and robustness. Therefore, knowing the accuracy of the global methodology presented is necessary to define the tolerances between the screwing tool and the fastener. The theoretical values provide a first approximation of the required accuracy; however, in a real application, there are more factors to take into account, such as metal dilation and tooling tolerance, etc.

### 4.1. Theoretical Tolerance

Groove angles determine the rotation tolerance ψ when the fastener turns inside the socket wrench, as shown in Figure 2a. Appendix A describes the mathematical analysis performed to determine the theoretical rotation and translation tolerances.

The rotation tolerance ψ is a function of the groove angle, based on a patent [21,22,23,24] and ISO tolerances [17,18,19,20]. Whereas, the translation error Δwx,Δwy is defined by the minimum and maximum ISO tolerances between the socket wrench and the fastener, also between the square drive and the coupler, as shown in Figure 2b.

### 4.2. Experimental Tolerance

The translation and rotation tolerances were obtained using a collaborative robot, a customized tool with various 12-point sockets, and a test bench in a perpendicular position with different fasteners sizes (M6, M8, M10, M16, M24), as shown in Figure 3. For each fastener, an initial rotation and translation position were defined for the correct insertion. Subsequently, several points were created around the initial position to program them into the robot, and then the movements were executed to determine the tolerances. For the translation, the M6, M8, and M10 used a combination of points in the range [−3 : 3] with a step of 0.25 mm (625 points), whereas, the M16 and M24 points in the range of [−6 : 6] with a step of 0.5 mm (576 points).The rotations used a range of [−16 : 16] with a step of 1 degree (66 points). The robot was set with a speed of 20 [mm/s] and an acceleration of 200 [mm/s${}^{2}$].

The data of position and forces exerted by the robot were sent to the virtual PLC using the Profinet protocol communication and saved into a datalogger for post-processing. When other fastener size is required in the test , the socket must be changed manually.

### 4.3. Results

The experimental translation tolerance represents the zones in which the socket wrench is inserted into the fastener. It corresponds to a workspace where the robot exerts a minimal insertion force. Figure 4 shows the results of the M6 fastener experiment. The blue and green zones are the regions where the socket is inserted correctly. The yellow and orange zones present locations where the socket is inserted by compliance and the robot exerts a higher force. The red zone shows the locations where the socket cannot be inserted, and the robot detects a collision force. The experiments show areas with some asymmetries due to the experiment’s uncertainties. For this reason, the tolerance has been defined as the radius of the maximum circle inside the blue area. Experiments were carried out with a robot nominal speed of 20 [mm/s] and an acceleration of 200 [mm/s${}^{2}$]. Other force values can be obtained by modifying the nominal values.

Regarding rotation tolerance values, Figure 5 shows the rotation results for the different fastener sizes (M6, M8, M10, M16, and M24). In this case, proper insertion and insertion by compliance are considered together; therefore, the results are binary, that is, the insertion right or the insertion is wrong.

Table 2 shows the theoretical and experimental results of the insertion experiments. The minimum and maximum translation and rotation tolerances values are obtained from the equations described in the Appendix A, and the maximum experimental tolerance values are obtained from the insertion experiments using the test bench shown in Figure 3. The standard column represents the theoretical rotation tolerance when the socket wrench does not have groove angles ψ=0 (Appendix A) (the first patents of socket wrenches). Additionally, the force exerted to perform the test properly are presented for several fastener sizes.

## 5. Performance of the Segmentation Neural Network (SNN)

The following models were trained using transfer learning and fine-tuning [56], the best model was selected by the EarlyStoppingPoint criterion [57]. The dataset consists of M10 and M8 fastener sizes distributed in 978 images to train and 416 to test. The hardware used is of great importance to reduce the time it takes for masks to be obtained from SNNs. In general, industrial PCs have no powerful CPU or GPU card. Therefore, an optimization process using embedded devices is necessary. The SNNs were tested on different devices, platforms, and variable optimization types (FP16, FP32, and UINT8) to determine the fastest related to FPS and the most accurate model through the MaP IOU ([58]).

Unet [41]. Devices PC Intel I9 and Nvidia GPU Gtx 1660 Ti.Mask RCNN [42]; backbone Resnet50. Devices PC Intel I9 and Nvidia GPU Gtx 1660 Ti.Deeplab [43]; backbone Mobilenet-v2, and EdgeMobilenet-1.0.Devices PC Intel I9, Nvidia GPU Gtx 1660 Ti, and Intel UHD Graphics 630.Embedded devices INCS2 [52], DETPU [53], and JJNX [51], respectively, into their optimization platforms Tensor RT 7.1.3 in Ubuntu 16.04; Openvino 2022.1.0.643 in Windows 10; and Google Coral Edge TPU Compiler 16.0.384591198 in Ubuntu 16.04.

Additionally, to compensate for possible changes in focus and brightness and improve the neural network, the image augmentation technique [59] with the python3.7 library “albumentations 1.0.0” was used with pixel-level transformations (GaussianBlur, MedianBlur, MotionBlur, AdditiveGaussianNoise, RandomBrightnessContrast, and RandomGamma) and spatial-level transformations (VerticalFlip, Rotate with interpolation, and HorizontalFlip).

### Results

Unet could be trained with few images and it does not require the use of transfer learning. The mask obtained was very well-defined, and the vertex could be obtained by a common-contour approximation algorithm [55]. Nevertheless, when using an image with different illumination, the mask is not perfect, as shown in Figure 6a.

Mask RCNN was trained with the Imagenet dataset and transfer learning. It has more robustness than Unet when new images with different illumination conditions and different hues of fasteners are used. However, the masks present ripples in their borders (Figure 6b) and is more difficult detecting a vertex using traditional vision algorithms [55,60].

Deeplab MovilNetV2 was trained with a VOC 2012 dataset and transfer learning. The mask obtained has border distortions, as shown in Figure 6c, in this case the use of traditional vision algorithms could not provide good performance. nevertheless, this SNN is faster than the others and can be optimized in other formats to reduce the inference time and work in real-time.

The previous analysis of the SNNs tested revealed that, in some cases, the masks obtained are not perfect; therefore, to increase the accuracy and robustness to obtain vertices and the center, the use of post-processing vision algorithms is necessary. In this work, a novel methodology and algorithms are proposed and described in the next section.

A comparison of the time inference and the MaP IOU performance of the previous SNN is presented in Table 3. The most accurate model, 98.97% Map IOU, is Deeplab EdgeMobilenet FP32, and the fastest 88.49 FPS is Deeplab Mobilenet-v2 FP32 using an Nvidia Gtx 1660 Ti 6 Gb GPU, and 26.38 FPS using an Intel Core I9 10-Gen-900H CPU. As a result, the best candidate is Deeplab, which has good inference time and precision. Therefore, Deeplab is used to calculate the performance of the algorithm proposed in the next section.

Table 4 presents a complete overview of the available hardware and performance. In general, when the model is optimized into a specified platform using other types of variables (FP16, UINT8), the precision is not reduced. Therefore, the use of each specific hardware depends on the application and the inference time necessary. The most accurate model, 98.93% MaP IOU, is Deeplab EdgeMobilenet UINT8, which was optimized in the Google Coral Edge TPU using the pre-quantization and running on DETPU, obtaining a time inference of 15.74 FPS. The fasted model, 50.54 FPS, is Deeplab Mobilenet-v2 FP16, optimized in OpenVino and running in Intel UHD Graphics 630. Deeplab Mobilenet, optimized in OpenVino, runs faster at 43 FPS on an Intel CPU and 50 FPS GPU, but running on INCS2 only obtains 6 FPS. Deeplab Mobilenet optimized in Google Coral Edge TPU can run in DETPU until 15 FPS using a pre-quantization process. The post-quantization process does not present good performance, with 1 FPS. Deeplab Mobilenet optimized in Tensor RT can run in JJNX until 25 FPS.

## 6. Algorithms Proposed to Obtain Vertices and Centers

### 6.1. Camera Parameters

Certain parameters must first be defined before working with different fastener sizes, such as the FOV, WD, pixel size, focus value, and exposure time.

In the first step, the FOV was obtained as a function of fastener size “*s*” and displacement tolerance Δh,Δw (Figure 7). The minimum fastener size to detect is related to the camera focus distance [75 : 1200] [mm], and the values are in the range of [10 : 160] [mm].

In the second step, the FOV was calculated using the pinhole camera model [25]; this model describes the geometry of the optical devices, and the FOV was obtained using triangle similarity. In this work, a variable fastener FOV was considered, where the tolerance Δh,Δw was set in millimeters. Whereas a constant FOV was defined by the ratio of the camera FOV and fastener FOV, as shown in Figure 7, the tolerance Δh was fixed at 249 pixels; hence, the FOV was 462 pixels, and the WD distance was calculated using Equation (Equation 1). The camera datasheet does not specify the sensor size (Sw,Sh), and these values were determined from the matrix calibration method ([32]), obtained as Sh=3.54 and Sw=4.72.

In the third step, the focus Equation (Equation 2) was obtained by taking several photos at different heights with the Omron autocalibration tool, and the pixel size Equation (Equation 3) was obtained by performing 16 camera calibrations at different heights with the Omron Multi-Dot Calibration tool, determining the ratio of pixels at 1 mm.

In the fourth step, the estimation of the exposure time value maintains constant brightness and contrast in the new photos. The algorithm implemented in [61] was modified to change the exposure time $t_{i}^{{}^{\prime }}$ and the ratio between the optimal average grayscale $g_{ref}$ and the average grayscale $g_{i}$ (Equation 4). The optimal average grayscale gref=84 was obtained from 400 pictures taken under the same exposure and illumination conditions.

The focus value and the exposure time are an approximation. Consequently, there may be a small difference in the image brightness and sharpness, which was compensated by the augmentation image technique. Meanwhile, the pixel approximation used an iterative algorithm, where the error between the center of the image and the center of the fastener has to be less than the corresponding error in Table 2.
(1)$$\begin{array}{cc} \; & \mathrm{Variable}\;\mathrm{FOV}\\ \; & WD_{\mathrm{mm}}=\frac{(S+2*\Delta h)*f}{Sh}\\ \; & \mathrm{Constant}\;\mathrm{FOV}\\ \;\; & r=\frac{FOVh}{FOV\_fh}=\frac{960}{462}=2.08\\ \; & WD_{\mathrm{mm}}=\frac{S\cdot r\cdot f}{S_{h}} \end{array}$$
(2)$$focus=0.8815*WD^{0.9953}$$
(3)$$pixelSize\left(1\;\mathrm{mm}\right)=3604.2*WD^{-0.971}$$
(4)$$if\left(\frac{\left|g_{i}-g_{ref}\right|}{g_{ref}}\right)\geq 10\%\to t_{i}^{{}^{\prime }}=t_{i}\frac{g_{ref}}{g_{i}}$$

### 6.2. Proposed Algorithms to Obtain Nut Center and Rotation

SNN masks are not always perfect; they may have ripples at their edges, and the vertices are not especially clear. Therefore, vision algorithms must be used to improve them. In this work, various algorithms are proposed to achieve the highest accuracy in the center and vertex rotation. Algorithm 1 summarizes the complete process, and Figure 8 shows each step.
**Algorithm 1:** Center&Vertices
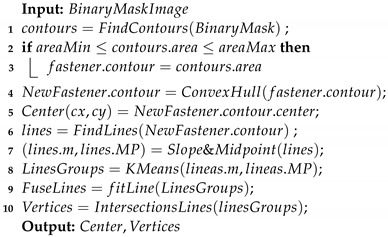


First, the algorithm receives the binary mask from the neural network inference (Figure 8a) and finds the contours using [45]. Sometimes, the image contains noise and detects small contours. The outlier contours are deleted by the hexagon area condition, which is defined as a function of the FOV and the fastener size (Equation 5) (Figure 8b). The convex hull algorithm [46] reduces edge ripples and border deformations. When the borders are correct, the center of the hexagon is obtained by contour moments (Figure 8c). The contour is transformed into several lines, applying the Hough transform [47] (Figure 8d), and K-means groups them into four classes using the middle point and slope as the criterion classification (Figure 8e). All lines in each group are merged into one single line using the least-squares method. Finally, the lines’ intersections correspond to possible vertices (Figure 8f).

When the mask is deformed, vertices do not correspond to the fastener hexagon; therefore, the voting algorithm [35] is modified and proposed to select the best vertices.

Algorithm 2 receives the vertices, the length of the fastener hexagon (Figure 7), and an admissible error. The length of the hexagon fastener (Equation 6) changes as a function of fastener size and FOV. The distance between the vertex candidate is calculated and compared with the distance between AB, AC, and AD. If the result is larger than the expected error, a vertex counter (vertex_cnt) increases and the vertex error (Terror) accumulates the values v1, v2, and v3. Finally, the best vertex is the lowest voting vertex, and the smallest amount accumulates errors.

Algorithm 3 calculates the rotation of the best vertex. First, the algorithm determines the vertex quadrant; next, the other five angles are then calculated by adding 60 degrees to the last one. Then, the angles in the first quadrant are selected. Finally, the angle with the lowest value with respect to the “x or y axis” is chosen as the angle rotation:(5)$$\begin{array}{cc} \; & area_{\min}=3\cdot {\left(s-error\right)}^{2}\cdot \tan\left(30{}^{\circ}\right)\cdot pixelSize\left(WD\right)\\ \; & area_{\max}=3\cdot {\left(s+error\right)}^{2}\cdot \tan\left(30{}^{\circ}\right)\cdot pixelSize\left(WD\right) \end{array}$$
(6)$$AB_{\mathrm{px}}=S\cdot \tan\left(30{}^{\circ}\right)\cdot pixelSize\left(WD\right)$$
**Algorithm 2:** BestVertex
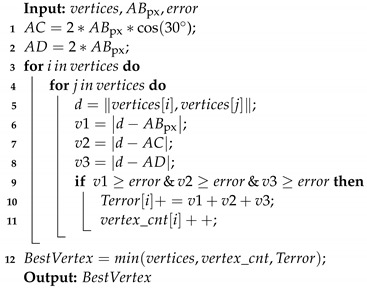


**Algorithm 3:** VertexRotation

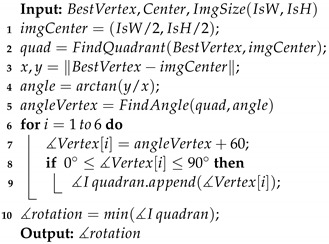



The test bed Figure 9 was built to analyze the performance of the algorithms proposed. The camera parameters were fixed according to the fastener size to work with a variable and constant FOV. Table 5 shows the values set using Equations (Equation 1)–(Equation 3), and Figure 10 shows a photo of each one.

The proposed algorithms were exhaustively tested with a large synthetic dataset (1,000,000 images) created from a real dataset (100 images or 10 images for each fastener size). These synthetic images seek to recreate several illumination conditions in the outside environment and the uncertainties in fastener positioning cause unfocused images. There were 500,000 images that were created with random brightness and sharpness and 500,000 images were created with brightness and sharpness controlled by the exposure algorithm (Equation 4). The input mask was obtained from the trained SNN Deeplab Mobilenet-v2 FP32 using a CPU I9. The highest available error is taken from the maximum theoretical and experimental error described in Section 4 and presented in Table 2. Additionally, the contour approximation algorithm [55] was tested with the same SNN mask and Algorithms 2 and 3 to compare the rotation results.

For each fastener size, 10 photos with different positions and rotations are taken, and vertices and centers are obtained manually with an image edition program (GIMP). Subsequently, the rotation angle was calculated using the average value of all the rotation angles from each vertex, and a similar process was performed related to the center. These values were used as the ground truth for each image. Finally, for each image, 10,000 synthetic images with several combination levels of brightness, unfocused, and sharpness were created using an image augmentation library [62]. The augmentations used were Blur, GaussNoise, HueSaturationValue, RandomBrightnessContrast, RandomGamma, and MotionBlur. Some synthetic images are presented in Figure 11.

Additionally, the methodology proposed was tested 616 times in real conditions of illumination and uncertain position and rotation in the test bench presented in Figure 12, using an M10 fastener as a test example. The tests involve positioning the camera at a random location (x,y) within the calculated FOV using the robot; in addition, the servomotor Qrob70 introduces an uncertain rz rotation between [0° : 360°], and the linear actuator introduces an uncertain rx rotation in the range of [0° : 45°]. The rx rotation is measured by an IMU and sent to the robot to correct the orientation.

The algorithm proposed then calculates the center and sends the data to the robot. It corrects the offset location until the center of the camera is the same as the fastener; then, the robot corrects the tool rotation and fits the socket wrench into the fastener. In this test, the correct insertion is verified using the force calculated experimentally in the previous section according to Table 2; the maximum force to insert correctly the socket into the fastener M10 is 24 N.

### 6.3. Results

Tests were performed using the SNN Deeplab Mobilenet-v2 FP32 on an Intel Core I9 10-Gen-900H CPU. The SNN time inference is 37.91 ms, whereas the time consumption required to execute the Algorithms 1–3 is 29 ms. Taking and downloading a picture takes approximately 650 ms. Furthermore, the algorithm to check the exposure time needs 7 ms in each iteration. Therefore, the execution time of one iteration is 723.91 ms. Figure 13 provides a first approximation of the performance using the proposed algorithm, which improved the vertex and center detection. The convex hull technique reduces the ripples and mask deformation, but some fuse lines selected by Kmeans classification could be wrong, such as Figure 13h, and intersecting lines may produce fake vertices; however, the proposed algorithm was designed to prevent these outliers and choose the best vertex. Therefore, the proposed methodology is a robust solution to tackle distorted masks present in unstructured illumination and different fasteners in the dataset.

First, the synthetic dataset with random brightness, unfocused, sharpness, and variable FOV was analyzed, using the maximum permissible tolerances shown in Table 2. The results are presented in Table 6. Concerning experimental tolerances values, the proposed algorithm has an efficiency of 89.90%, with an average error value of −0.36° and a standard variation of 1.91°, whereas the center has an efficiency of 99.99%, with an average error value of 0.14 mm and a standard variation of 0.06 mm. Regarding theoretical tolerances values, the rotation detection has an efficiency of 92.56%, with an average error value of 0.13° efficiency of 99.99%, an average error value of 0.14 mm, and a standard variation of 0.06 mm.

Second, photos with a constant fastener FOV were taken. The exposure algorithm controlled the maximum and minimum brightness and sharpness when a new synthetic image was created. In this case, the exposure condition (Equation 4) checks if the average pixel value is lower than the reference pixel value; if not, a new synthetic image was tested.

This test simulated the stability of the control of exposure time to capture light in the camera sensor. Figure 14 shows a sample with a real camera applying the exposure algorithm. In the first step, the image has oversaturated lighting, and the new exposure time is then calculated to take a new picture, and the process is repeated until the condition is satisfied. In the test performed with the real camera, the algorithms calculate the new time exposure in three iterations (2.2 s). Additionally, to conduct a comparative performance with the proposed algorithm, the algorithm of contour approximation [55] is tested, with the same mask from the SNN as input.

Table 7 shows the data collected in the test, where the results show a notable increase in performance. The accuracy with the experimental rotation tolerance value is 99.86%, and the theoretical tolerance value is 99.91%, whereas, with the experimental and theoretical translation tolerance, the accuracy is 100%.

The algorithm of contour approximation shows a lower accuracy (15.16%) using the experimental rotation tolerance. These results clearly demonstrate that the proposed algorithm has high accuracy and robustness.

Third, the test in real conditions demonstrates that the proposed algorithms implemented work exceptionally well in an unstructured and illuminated environment; the accuracy (96.26%) is high, and the errors are minimal. Table 8 shows the results, where the average number of steps to match the camera with the center of the fastener is 2.08 iterations, while the time required is 2.49 s. This time could be improved with higher robot speed and the use of another interface to download the photo from an industrial camera. The complete process of the real tests is presented in the video.

## 7. Conclusions

The analysis of the data presented in Table 2 reveals that a socket wrench designed with groove angles increases the theoretical rotation tolerance by an average of 6.67 degrees compared to a standard socket wrench without groove angles. Additionally, some patents could have a slightly higher tolerance, such as patent 2 [22], with an average value of 0.64 degrees, with respect to other patents [21,23,24].

The translation tolerances increase with higher sizes of fasteners from 0.75 for M6 to 2.50 for M24; however, the rotation decreases with the size, from 5.5° for M6 to 3.5° for M24. In other words, with higher fastener sizes, the algorithm used to detect the center could have more tolerance, but the algorithm used to detect the rotation has to be more accurate. The opposite logic holds when working with smaller fastener sizes.

When the fastener size is higher, the force exerted by the robot increases because the insertion distance between the socket wrench and the fastener is larger and there is more friction.

The values of theoretical and experimental tolerances are not similar, but are close due to other factors not considered in the analysis, such as fastener and socket wrench manufacturing processes, metal dilatation, tooling tolerance, etc. However, the use of the theoretical tolerances gives a first approximation of the application performance and the algorithms developed, but the experimental tolerances show real accuracy.

SNNs work very well in harsh environments and various lighting conditions; however, the SNNs tested reveal the masks obtained in some cases are not perfect and require additional post-processing vision algorithms to increase the accuracy and robustness to use in robotic applications where vertices and the center require high precision.

The Deeplab model exhibits good performance in relation to Unet and Mask RCNN. The most accurate model, with 98.97% Map IOU, is Deeplab EdgeMobilenet FP32, and the fastest, with 88.49 FPS, is Deeplab Mobilenet-v2 FP32 running in Nvidia Gtx 1660 Ti 6 Gb GPU. The slower SNN was Mask RCNN, with 0.38 FPS. Regarding embedded devices, the accuracy of the model is not affected (less than 1%) when it is optimized in FP32, FP16, and UINT8 using a specified platform. Therefore, the use of the specific hardware described depends on the application and the inference time required. Deeplab EdgeMobilenet UINT8 is the most accurate model (98.93% MaP IOU, 15.74 FPS), and it was optimized in Google Coral Edge TPU using the pre-quantization and running on DETPU. However, the post-quantization process does not present a good performance with 1 FPS. A Deeplab model optimized in OpenVino and running in Intel UHD Graphics can reach 50.54 FPS, while running on INCS2 only obtains an FPS of 6. Deeplab Mobilenet optimized in Tensor RT can run in JJNX until 25 FPS. However, its device is 10 times more expensive than DETPU and INCS2.

The proposed methodology is a robust solution to tackle outlier contours and fake vertices produced by distorted masks present in unstructured illumination and worn, scratched, and rusty fasteners. Process time consumption has been registered in an Intel Core I9 10-Gen-900H CPU, where the SNN time inference is 37.91 ms, the Algorithms 1–3, (Equation 4) are executed in 36 ms, and a picture is taken and downloaded in 650 ms, for a total of 723 ms.

The results based on experimental tolerances and a synthetic dataset with random brightness, unfocused, sharpness, and variable FOV show a rotation accuracy of 89.90%, an average error of −0.36°, a center detection of 99.99%, and an average error value of 0.14 mm.

The results using experimental tolerance with a synthetic dataset, constant FOV, and constant pixel intensity (Equation 4) show a rotation with an accuracy of 99.86% and an average error of 0.1°, whereas the accuracy of center detection is 100%.

This means that the camera exposure time as a function of the average pixel intensity from the previous dataset and a constant FOV significantly increases (10%) the performance of the application with respect to a variable FOV and pixel intensity. Additionally, the algorithm of contour approximation tested with the same synthetic images using constant FOV and pixel intensity shows a lower accuracy (15.16%) using the experimental rotation tolerance. These results clearly demonstrate that the proposed algorithm has high accuracy and robustness. The 616 tests in a real environment and various illumination conditions demonstrate that the proposed algorithms work exceptionally well with an accuracy of 96.26%.

The use of force sensors increases the reliability of the process by providing information on the status of the task and effective feedback for automatic error correction. The values of the maximum forces exerted by the robot when fitting the socket wrench into various fastener sizes are different, which means that to check the proper fit operation, it is necessary to use a correct value of force.

In future work, to increase the robustness of task supervision, fixed position feedback could be used; for example, if the force and position values exceed or do not reach an established limit, a collision or success status can be defined. However, it works well on non-deformable surfaces. To work on any surface, new sensors and customized tools could be used, such as load cells, which obtain the force of the spring mounted on the transmission axis when the socket wrench fits the fastener. Additionally, the use of deep (RGBD) or stereo cameras could be considered to determine the fastener orientation on non-perpendicular surfaces and extend the capability of the proposed methodologies to work in any environment.

This contribution has great value for robotic screwing tasks. It presents a complete guide and factors to be considered in new developments.

## Figures and Tables

**Figure 1 sensors-23-04527-f001:**
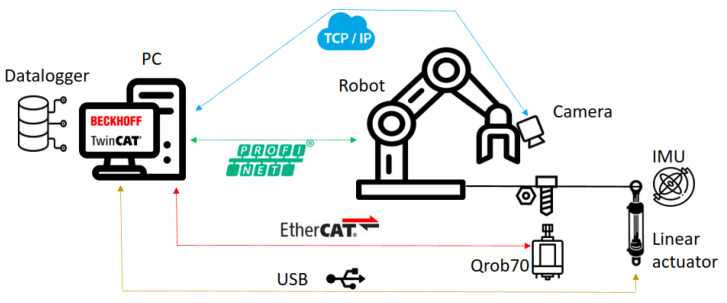
General testing setup used to determine the experimental rotation and translation tolerances, which also carried out the performance analysis of the algorithms proposed. The PC works as a master and a datalogger for controlling and receiving data from the collaborative robot and the camera. The servomotor Qrob70, linear actuators, and IMU are used to introduce and read random values in the rz and rx axis with respect to the robot axis when real tests are performed.

**Figure 2 sensors-23-04527-f002:**
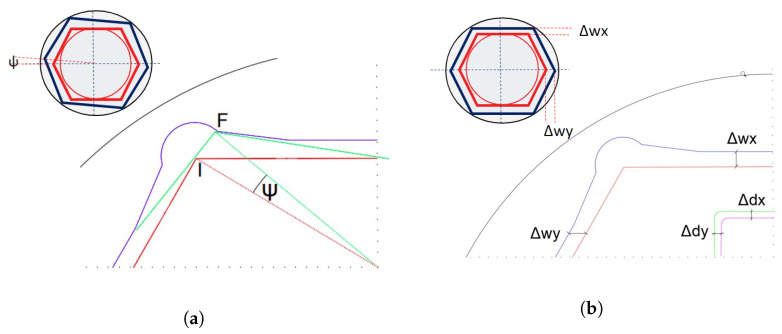
Rotation and translation tolerances between a fastener and socket wrench. The complete analysis of parameters is described in the Appendix A. (**a**) Theoretical rotation tolerance ψ as functions of ISO [17,18,19,20] and patents [21,22,23,24] manufactured when the fastener turns from the point I (red) to point F (green) inside the socket wrench (black) with groove angle (blue). (**b**) Theoretical translation tolerance Δx=Δwx+Δdx,Δy=Δwy+Δdy as functions of ISO tolerances inside of the socket wrench (black) [17,18,19,20] of fastener (red) with socket (blue), and (green) square drive with (purple) square coupling (green).

**Figure 3 sensors-23-04527-f003:**
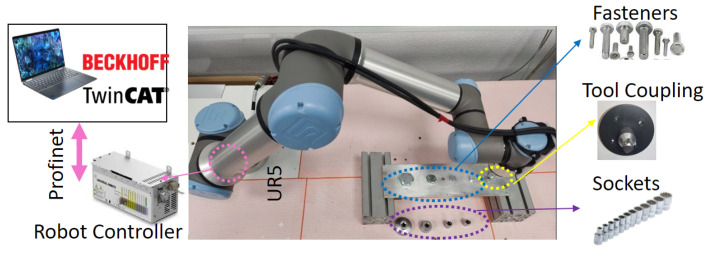
Architecture to determine the experimental translation and rotation tolerances in variuos fastener sizes (M6, M8, M10, M16, and M24). The robot moves several points in Cartesian coordinates (x,y) and a rotation (rz) around the fastener to determine when the tool can be inserted. Position and force data from the collaborative robot are sent by Profinet to a datalogger implemented in a virtual Beckhoff PLC mounted in a PC.

**Figure 4 sensors-23-04527-f004:**
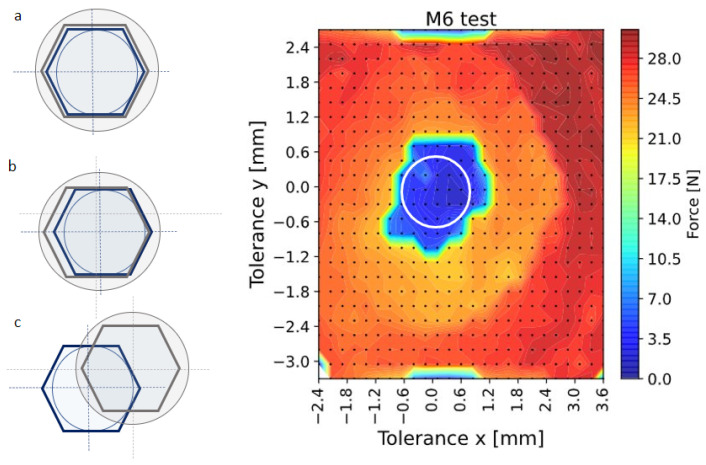
Experimental study of the translation tolerances between the center of the M6 fastener and the socket wrench. There are three possible cases: (**a**) proper insertion, interaction force less than 14 N (blue); (**b**) insertion with compliance, interaction force between 14 and 21 N (orange); and (**c**) failed insertion higher than 21 N (red). As result, proper insertion is achieved with a tolerance of ±0.75 mm.

**Figure 5 sensors-23-04527-f005:**
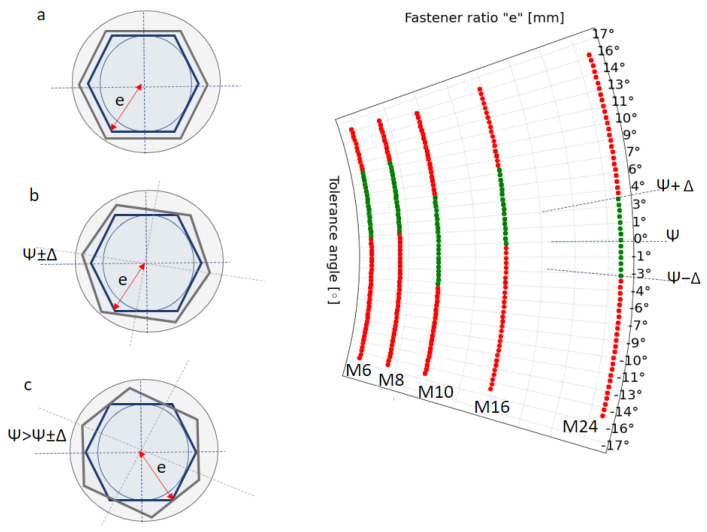
Experimental study of rotation tolerances using fastener metrics (M6, M8, M10, M16, and M24). The curve lines represent the result of tests carried out when the robot rotated the socket wrench to a certain degree around the fastener. There are two possible cases: (**a**,**b**) proper insertion (green), (**c**) failed insertion (red). The rotation tolerance is the STD value ψ±Δ around the average value ψ.

**Figure 6 sensors-23-04527-f006:**
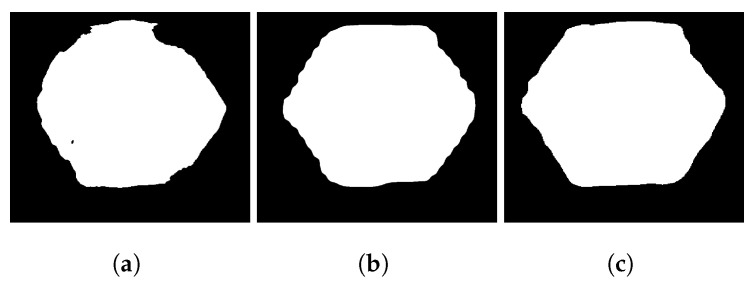
Distorted inference masks obtained from the SNN tested. (**a**) Unet, for which the mask is not particularly well-defined when new data are presented for detection. (**b**) Mask RCNN presents ripples in the edges; however, it is more robust to new fastener images. (**c**) Deeplab MovilNetV2 presents deformations in the mask.

**Figure 7 sensors-23-04527-f007:**
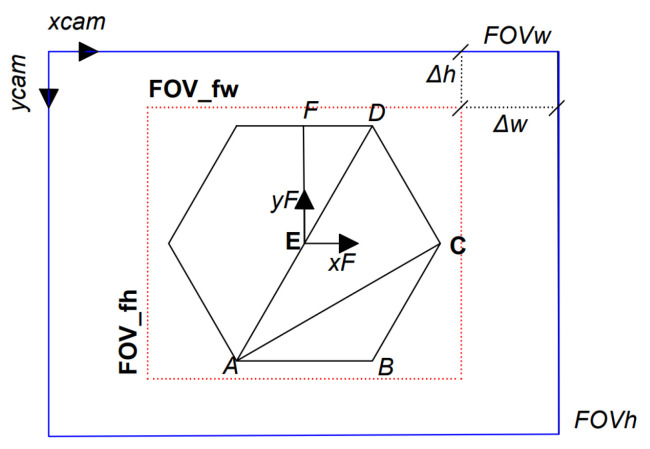
Definitions of parameters used in a fastener photo. Camera reference xcam,ycam; camera field of view FOVh,FOVw (blue). Fastener field of view FOV_fh,FOV_fw (red). Δh,Δw displacement tolerance. Distance between two fastener vertices, AB, AC, AD. Distance from center to vertex AE. Fastener size s=AC=2∗EF.

**Figure 8 sensors-23-04527-f008:**
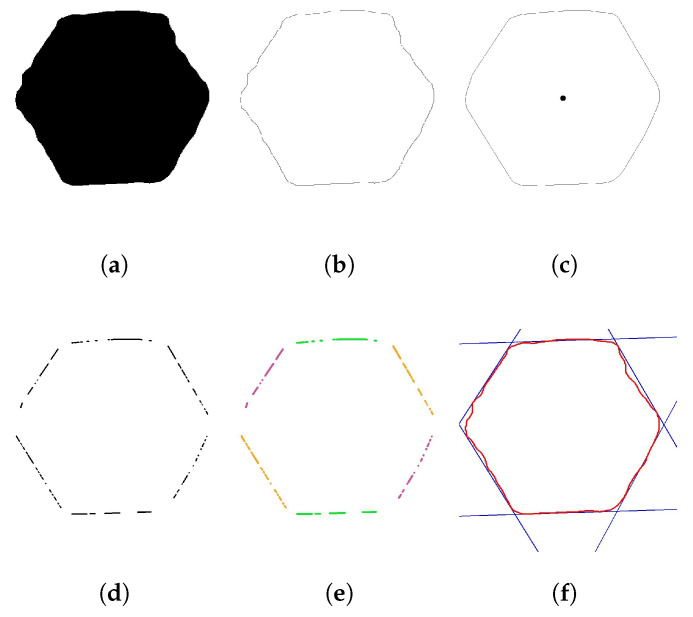
Proposed methodology described in the Algorithm 1 to obtain the center and vertices. (**a**) Obtain a mask from SNN. (**b**) Find contours and remove outliers contours. (**c**) Convex hull contour and obtain the center. (**d**) Convert contour to lines. (**e**) Clustering lines. (**f**) Fuse groups of lines and obtain the intersection between them to obtain potential vertices.

**Figure 9 sensors-23-04527-f009:**
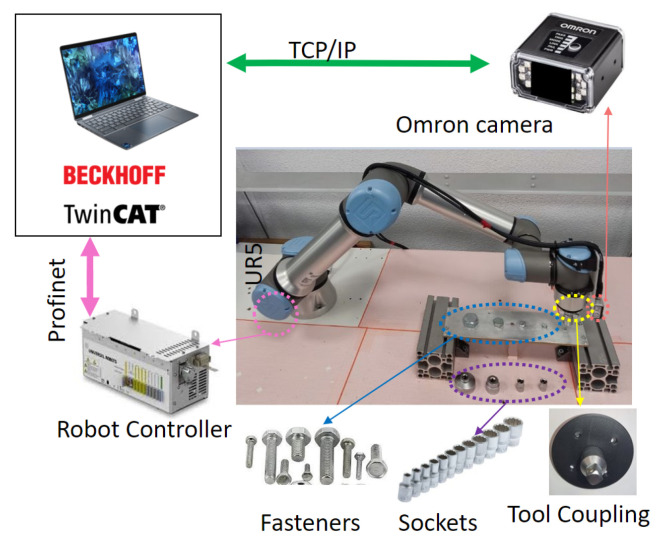
The architecture used to take photos of various fastener sizes (M6, M8, M10, M16, and M24) with fixed and variable FOV. FOV, focus, pixel size, and exposure were determined from Equations (Equation 1), (Equation 2), (Equation 3), and (Equation 4), respectively, and the values are shown in Table 5.

**Figure 10 sensors-23-04527-f010:**
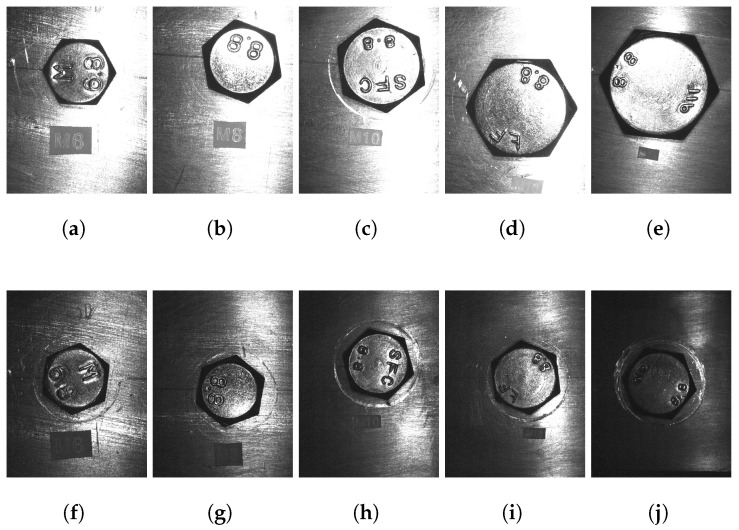
Some fastener photos used to test the algorithm proposed. The upper photos show a variable FOV defined by a tolerance in millimeters, whereas the bottom photos show a constant FOV defined by a variable tolerance in pixels. Photos with variable FOV (Equation 1) (**a**) M6, (**b**) M8, (**c**) M10, (**d**) M16, and (**e**) M24. Photos with constant FOV (Equation 1) (**f**) M6, (**g**) M8, (**h**) M10, (**i**) M16, and (**j**) M24.

**Figure 11 sensors-23-04527-f011:**
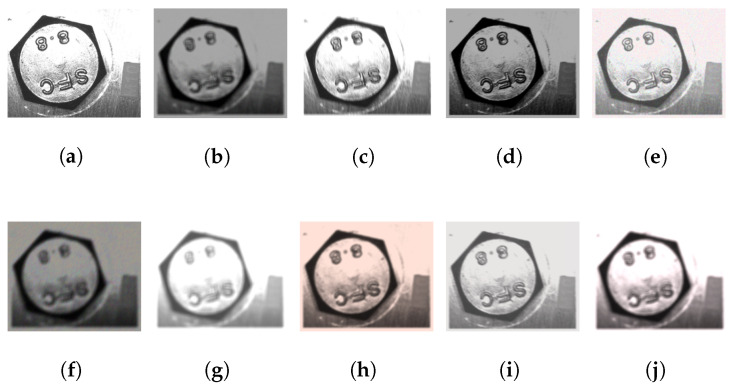
Random examples created by the augmentation image algorithm in the synthetic dataset. (**a**) Original image. (**b**–**j**) Images created using a combination of the augmentation Blur, GaussNoise, HueSaturationValue, RandomBrightnessContrast, RandomGamma, and MotionBlur.

**Figure 12 sensors-23-04527-f012:**
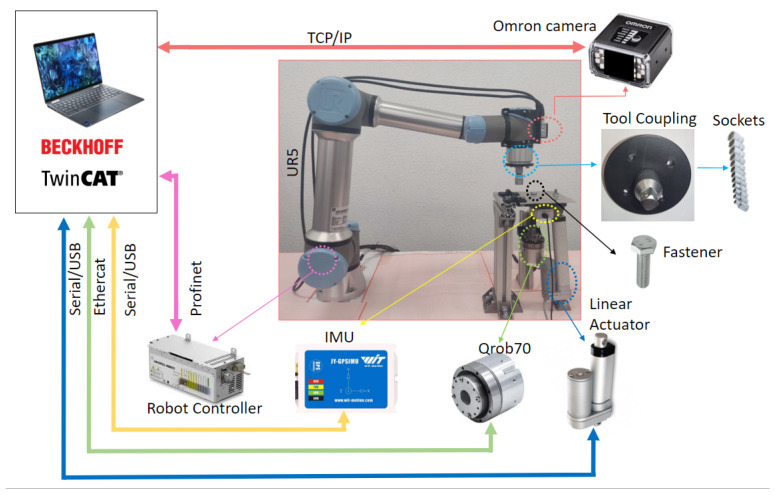
Test bed to execute a real example application using an M10 fastener. The robot starts in random positions (x,y) and the servomotor Qrob70 rotates the fastener an rz random turn; a linear actuator creates uncertain positions in rx based on a robot base coordinate, and the rx rotation is measured using an IMU. The force of the robot is measured to confirm that the task has been carried out properly. The video shows some tests in different illumination conditions.

**Figure 13 sensors-23-04527-f013:**
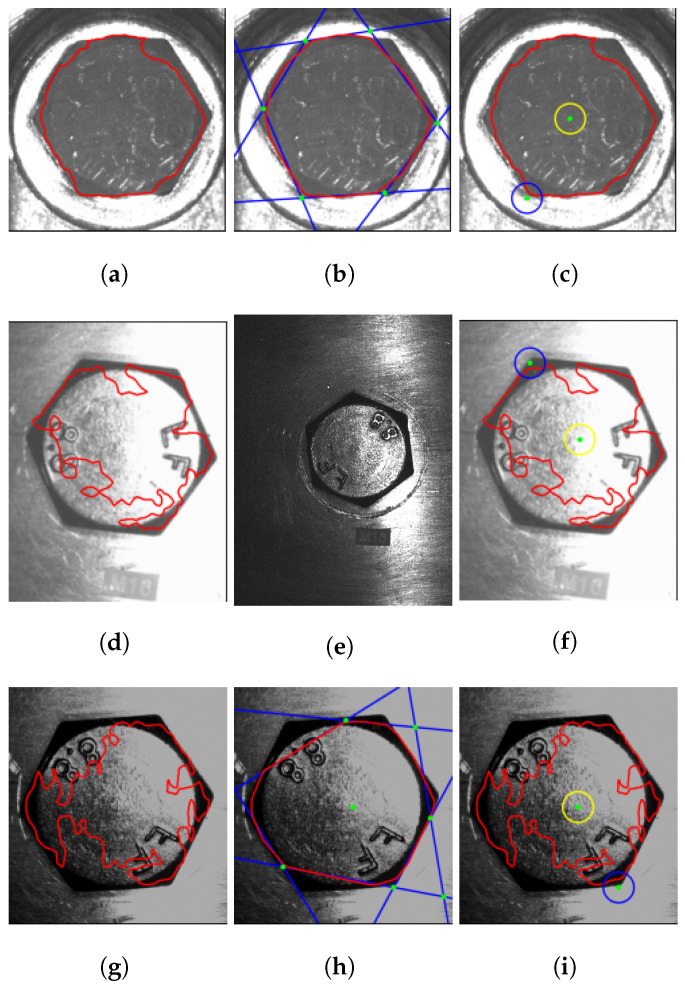
Tests using the proposed algorithm when the SNN cannot obtain defined masks. The best vertex and center with a deformed mask can be detected when different fasteners and illumination are used. (**a**–**c**) Test 1 with an M19 fastener. (**d**–**f**) Test 2 with an M16 fastener. (**g**–**i**) Test 3 with an M16 fastener. The left-hand images are the original image with the mask inferred by the SNN (red). The middle images are the mask improved with the convex hull technique (red). Intersection lines (blues) after the edges are transformed into lines and are clustered in groups. Vertices formed at the intersection of the lines (green). The right-hand images are the best vertex (blue-green) and the center (yellow-green) detected.

**Figure 14 sensors-23-04527-f014:**
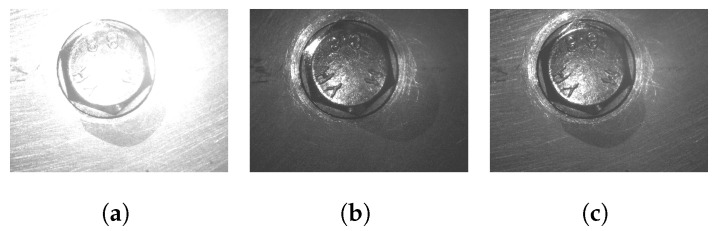
Brightness correction using the auto-exposure condition (Equation 4) with a 40,202 lux illumination environment. In the beginning, the camera took a photo to check the actual average pixels, after which some iterations were required to obtain the average grayscale (84), which was calculated from previous images in the dataset. (**a**) Iteration 1, 206 average pixel value. (**b**) Iteration 2, 103-pixel value. (**c**) Iteration 3, 88-pixel value.

**Table 1 sensors-23-04527-t001:** Advantages and disadvantages of vision and force sensors in (un)screwing tasks.

Sensor	Advantages	Disadvantages
Vision	•Vision provides image feedback on the process and allows the use of several vision algorithms and neural networks to detect fasteners. •The use of filters in a determinate wavelength range could improve detection. •The designed algorithms can be tested using synthetic images simulating several illumination conditions. •Depending on the camera resolution, the precision can be high.	•Depending on the environment, the illumination could not be constant. •In small workspaces, the use of external illumination is not possible. •The fixed lens and focus values only allow work with a determined field of view (FOV); consequently, it can only work within a determinate range of fastener sizes •Robot vibration or deceleration can interfere while the images are taken, causing blur.
Force	•Force feedback provides the task knowledge of the surface environments and can be used in a master–slave architecture to use in teleoperation tasks. •Several types of control based on the force can be used, such as control based on active compliance to search the fastener or detect collisions. •The use of collaborative robots with integrated force sensors are cost-efficient and adaptable to different manufacturing scenarios.	•The search time of active compliance is inversely proportional to the fastener’s size and can be time-consuming. •In some cases, the estimated center of the screw should be taught to the robot by a human operator. •High-precision force sensors are expensive. •Friction between the screw head surface and the tool, a stuck screw, and a damaged screw head affects the algorithm’s performance.
Vision–Force	•The vision provides an exact approximation of the object and the force feedback ensures safety in the process of operation.	•The cost is more expensive. •Software skills are more extensive.

**Table 2 sensors-23-04527-t002:** Minimum and maximum translation and rotation tolerances values obtained from theoretical equations described in Appendix A, and the maximum experimental tolerances values obtained when the robot moved several points in Cartesian coordinates and z-axis rotation using the test bed shown in Figure 3. The standard column represents the theoretical rotation tolerance when the socket wrench does not have groove angles ψ=0 (Appendix A) (first patents of socket wrenches). The P1, P2, and P3 columns represent the theoretical rotation values as a function of the socket wrench patent used. The force experimental column represents the maximum force value when the socket wrench is inserted correctly into the fastener, as described in Figure 4.

Metric	Translation Tolerance (mm)		Rotation Tolerance (°)					Force Experimental max (N)
	Theoretical [min max]	Experimental max	Theoretical [min max]				Experimental max	
			Standar	P1 [21]	P2 [22]	P3 [23,24]		
M6	[0.07 : 0.74]	0.75	[0.04 : 0.12]	[3.31 : 7.43]	[3.87 : 7.96]	[2.99 : 7.25]	5.50	14
M8	[0.08 : 0.82]	0.88	[0.04 : 0.11]	[3.34 : 7.11]	[3.90 : 7.64]	[3.03 : 6.92]	4.75	19
M10	[0.09 : 0.87]	1.13	[0.04 : 0.11]	[3.36 : 6.67]	[3.91 : 7.21]	[3.05 : 6.46]	4.25	24
M16	[0.10 : 1.04]	2.50	[0.04 : 0.10]	[3.32 : 6.24]	[3.87 : 6.77]	[3.00 : 6.01]	4.00	19
M24	[0.11 : 1.44]	2.50	[0.04 : 0.10]	[3.28 : 6.45]	[3.83 : 6.98]	[2.96 : 6.23]	3.50	38

**Table 3 sensors-23-04527-t003:** Comparison of the semantic neural network (SNN) using an Intel Core I9 10-Gen-900H CPU and the Nvidia Gtx 1660 Ti 6 Gb GPU. The weight column represents the size of the file. The MaP IoU column is the mean average precision using the intersection over the union. In addition, FPS values were calculated to determine the performance between them.

Neural Network	Weight	MaP IOU	FPS	
	Mb		I9	GPU
Unet FP32	355.00	97.45	0.44	12.20
Mask RCnn resnet50 FP32	170.00	98.26	0.38	14.70
Deeplab Mobilenet-v2 FP32	8.54	98.84	26.38	88.49
Deeplab EdgeMobilenet FP32	14.42	98.97	21.88	81.30

**Table 4 sensors-23-04527-t004:** Performance of Deeplab versions using different types of format variables in GPU, CPU, and embedded devices. Model precision, MaP IOU, and FPS were obtained from the tests performed on Intel UHD Graphics 630 (Intel GPU), CPU Intel Core I9 10-Gen900H (Intel I9 CPU), Nvidia GTX 1660 super 6 Gb (Gtx GPU), and embedded devices (JJNX, INCS2, DETPU) into their respective optimization platforms (Tensor RT, Openvino, Google Coral Edge TPU). DETPU was tested using a quantization process POSTQ and PREQ.

OpenVino	Intel I9 CPU		Intel GPU		INCS2	
	MaP IOU	FPS	MaP IOU	FPS	MaP IOU	FPS
Deeplab Mobilenet-v2 FP16	98.78	43.29	98.78	50.54	98.78	6.07
Deeplab Mobilenet-v2 FP32	98.82	42.55	98.82	33.34	98.82	6.07
**Google Coral Edge TPU**	**DETPU POSTQ**		**DETPU PREQ**			
	MaP IOU	FPS	MaP IOU	FPS		
Deeplab Mobilenet-v2 UINT8	98.78	1.12	98.64	15.36		
Deeplab EdgeMobilenet UINT8	98.59	1.12	98.93	15.74		
**Tensor RT**	**JJNX**					
	MaP IOU	FPS				
Deeplab Mobilenet-v2 FP16	98.80	25.06				
Deeplab Mobilenet-v2 FP32	98.84	12.19				

**Table 5 sensors-23-04527-t005:** Camera settings defined by Equations (Equation 1)–(Equation 4) to work with different fastener sizes. The variable FOV is defined by tolerance Δh,Δw in millimeters, and the constant fastener FOV is defined by a tolerance Δh,Δw in pixels (Figure 7). The pixel size value is the number of pixels in one millimeter.

	Variable Fastener FOV			Constant Fastener FOV		
Metric	WD (mm)	Focus	Pixel Size	WD (mm)	Focus	Pixel Size
M6	90.39	78.00	45.44	94.00	81.00	43.70
M8	103.95	89.00	39.67	122.21	105.00	33.9
M10	122.03	105.00	33.95	159.82	138.00	26.13
M16	153.67	132.00	27.09	225.63	193.89	18.19
M26	207.90	178.00	20.24	338.44	290.28	12.61

**Table 6 sensors-23-04527-t006:** Accuracy results using the proposed algorithm to find the center and the rotation at various fastener sizes (M6, M8, M10, M16, and M24). A synthetic dataset was created from photos taken with a variable FOV (upper photos, Figure 10), and augmentation techniques as shown in Figure 11. The tolerance was obtained from the maximum theoretical and experimental values described in Table 2. The samples were 100,000 images per metric, 500,000 samples in total. Exp = experimental tolerance, P1 = theoretical patent1 tolerance [21], P2 = theoretical patent2 tolerance [22], and P3 = theoretical patents3 tolerance [23,24].

		Rotation (°)							Traslation (mm)					
		Ok			Wrong				Ok			Wrong		
Metric	Tolerance	Samples %	Mean	STD	Samples %	Mean	STD	Tolerance	Samples %	Mean	STD	Samples %	Mean	STD
**M6**	Exp < 5.5	91.13	−0.43	1.94	8.87	4.14	31.29	<0.75	100.00	0.12	0.05	0.00	0.00	0.00
	P1 < 7.43	99.15	−0.13	1.59	0.85	−43.53	17.18	<0.74	100.00	0.12	0.05	0.00	0.00	0.00
	P2 < 7.96	99.16	−0.13	1.59	0.84	−43.74	17.01							
	P3 < 7.25	99.15	−0.13	1.59	0.85	−43.4	17.28							
**M8**	Exp < 4.75	88.92	−0.36	1.81	11.08	2.75	28.22	<0.88	100.00	0.14	0.06	0.00	0.00	0.00
	P1 < 7.11	94.35	−0.56	2.20	5.66	9.12	38.2	<0.82	100.00	0.14	0.06	0.00	0.00	0.00
	P2 < 7.65	95.07	−0.61	2.28	4.93	11.36	40.39							
	P3 < 7.25	94.00	−0.55	2.17	6.00	8.25	37.25							
**M10**	Exp <4.25	94.82	−0.03	1.28	5.18	−12.02	35.41	<1.13	99.99	0.10	0.06	0.01	24.25	0.00
	P1 < 6.67	96.81	−0.05	1.48	3.42	−17.89	42.86	<0.87	99.99	0.10	0.06	0.01	24.25	0.00
	P2 < 7.21	96.98	−0.043	1.51	3.02	−20.32	44.27							
	P3 < 6.46	96.71	−0.04	1.47	0.03	−18.56	42.23							
**M16**	Exp < 4.00	89.90	−0.57	1.96	89.90	−0.57	1.96	<2.50	97.49	1.25	0.68	2.51	6.24	7.80
	P1 < 6.24	90.68	−0.61	2.81	9.32	−0.61	2.81	<1.04	96.96	0.64	0.22	3.03	2.83	4.44
	P2 < 6.77	90.92	−0.76	3.11	9.08	−0.76	3.11							
	P3 < 6.01	90.60	−0.61	2.72	9.40	−0.61	2.72							
**M24**	Exp < 3.50	87.51	0.40	1.91	12.49	−11.18	1.91	<2.77	99.73	1.55	0.68	0.27	4.17	1.60
	P1 < 6.45	89.27	0.85	3.46	10.73	−15.37	3.46	<1.44	95.01	0.77	0.43	2.51	3.37	1.66
	P2 < 6.98	89.57	0.87	3.70	10.43	−16.18	3.70							
	P3 < 6.23	89.14	0.86	3.32	10.86	−14.98	3.32							
**Total Theoretical**		**92.56**	**−0.13**	**1.95**	**7.44**	**−11.60**	**17.23**		**99.99**	**0.14**	**0.06**	**1.39**	**5.21**	**1.63**
**Total Experimental**		**89.90**	**−0.36**	**1.91**	**11.08**	**−0.57**	**28.22**		**99.99**	**0.14**	**0.06**	**0.27**	**6.24**	**1.60**

**Table 7 sensors-23-04527-t007:** Accuracy results using the proposed algorithm to find the center and rotation in various fastener sizes (M6, M8, M10, M16, and M24). A synthetic dataset was created from photos taken with constant FOV (bottom photos, Figure 10), augmentation techniques (as shown in Figure 11), and with controlled brightness and sharpness defined by the time exposure condition (Equation 4). The tolerance was obtained from the maximum theoretical and experimental values described in Table 2. Furthermore, the contour approximation algorithm [55] is tested to compare it with the proposed algorithm to obtain the rotation. The samples were 100,000 images per metric, 500,000 samples in total. Exp = experimental tolerance, Coa = contour approximation algorithm using the same value of Exp, The = theoretical patent1 tolerance [21].

		Rotation (°)							Traslation (mm)					
		Ok			Wrong				Ok			Wrong		
Metric	Tolerance	Samples %	Mean	STD	Samples %	Mean	STD	Tolerance	Samples %	Mean	STD	Samples %	Mean	STD
**M6**	Exp < 5.50	99.90	−0.17	1.11	0.10	25.82	5.43	Exp < 0.75	100	0.02	0.02	0.00	0.00	0.00
	CoA < 5.50	50.78	−0.21	2.19	49.22	6.26	9.35							
	The(pm) < 7.43	99.91	−0.17	1.11	0.10	26.25	4.63	The < 0.74	100.00	0.02	0.02	0.00	0.00	0.00
**M8**	Exp < 4.75	98.80	0.11	1.53	1.20	5.26	6.91	Exp < 0.88	100.00	0.03	0.02	0.00	0.00	0.00
	CoA < 4.75	31.05	−0.80	1.53	68.95	1.76	14.32							
	The < 7.11	99.66	0.13	1.60	0.34	11.98	6.59	The < 0.82	100.00	0.03	0.02	0.00	0.00	0.00
**M10**	Exp < 4.25	99.73	0.18	0.85	0.27	8.54	14.92	Exp < 1.13	100.00	0.03	0.03	0.00	0.00	0.00
	CoA < 4.25	15.16	−1.24	1.08	0.27	−9.02	7.07							
	The < 6.67	99.79	0.18	0.85	0.21	10.98	15.92	The < 0.87	100.00	0.03	0.03	0.00	0.00	0.00
**M16**	Exp < 4.00	99.97	−0.12	0.89	0.03	−4.68	1.98	Exp < 2.50	100.00	0.05	0.05	0.00	0.00	0.00
	CoA < 4.00	14.64	2.43	1.07	85.36	−1.22	14.68							
	The < 6.24	99.99	−0.12	0.89	0.01	-6.68	0.28	The < 1.04	100.00	0.05	0.05	0.00	0.00	0.00
**M24**	Exp < 3.50	99.86	0.01	0.95	0.14	7.00	7.90	Exp < 2.77	100.00	0.08	0.07	0.00	0.00	0.00
	CoA < 3.50	14.15	2.08	1.46	85.85	−0.79	14.40							
	The < 6.45	99.96	0.01	0.96	0.04	17.19	9.60	The < 1.44	100.00	0.08	0.07	0.00	0.00	0.00
**Total Theoretical**		**99.91**	**0.01**	**0.96**	**0.10**	**11.98**	**6.59**		**100.00**	**0.03**	**0.03**			
**Total Experimental**		**99.86**	**0.01**	**0.95**	**0.14**	**7.01**	**6.91**		**100.00**	**0.03**	**0.03**			
**Total Contour Aprox**		**15.16**	**−0.21**	**1.46**	**68.95**	**−0.79**	**14.32**							

**Table 8 sensors-23-04527-t008:** Performance of the proposed algorithms tested in the test bench (Figure 12) and in real environments under various illumination conditions (video). The correct column corresponds to the proper task execution, based on the measurement of the force and the maximum allowed value presented in Table 2. Average # of iterations corresponds to the number of steps to match the camera center and the fastener center and the time required.

Tests	Average # of Iterations	Average Time (s)	Correct	Incorrect	Accuracy %
616	2.08	2.49	593	23	96.26

## Data Availability

Not applicable.

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
