# Peer review of "Robust Fastener Detection Based on Force and Vision Algorithms in Robotic (Un)Screwing Applications"

_sensors, 2023, doi:10.3390/s23094527_

Round 1

Reviewer 1 Report

1. Please recheck the entire manuscript carefully:

- some misspelled words 

- Some manuscript formats were not consistency, please check the journal's author guidelines.

- both the decimal comma and decimal point appeared to be confused. 

- Table 3 Pre Q: Post-quantization => Pre-quantization

- Line No. 283: delta_h,w => delta_h, delta_w

- Line No. 297: equation (2) => Eq.(3)

- Line No. 316: Algorithm regalg:algorithm1 => Algorithm 1

- Line No. 318-326: Fig.10 => Fig.8

and some more...

2. Please explain why the mathematical analysis of translation and rotation errors was important to the proposed method and how they were related to the proposed work. Also, why did we need to compare the theoretical translation/rotation tolerance with the experimental tolerances?

3. Please explain why the results shown in Fig. 4 was important. Is it the common knowledge when a socket is inserted to a fastener? 

4. Please include some synthetic images created using the image augmentation library. 

Author Response

Reviewer 1

Sensors (ISSN 1424-8220)

Subject: Response to Reviewer 1 Comments

Thanks for the time to analyze our paper. We must express our gratitude for your insightful comments and suggestions made. The comments have served to improve and complement the research and emphasize the quality of presentation and dissemination of the work.

The comments were responded at the end of this letter, describing the changes we have made and explaining the related problems' relevant aspects. Additionally, the modified pdf file is added with the changes highlighted in various colors.

  1. Please recheck the entire manuscript carefully:

- some misspelled words 

An extensive linguistic revision of the article has been carried out, correcting several syntax and grammar errors in the manuscript.

- Some manuscript formats were not consistency, please check the journal's author guidelines.

   It has been detected some inconsistencies that have been modified.

- both the decimal comma and decimal point appeared to be confused. 

    In the current version, point is used for decimal only in text and tables.

- Table 3 Pre Q: Post-quantization => Pre-quantization

   Is done. Line 333.

- Line No. 283: delta_h,w => delta_h, delta_w

    Is done. Line 339.

- Line No. 297: equation (2) => Eq.(3)

     Is done. Line  350.

- Line No. 316: Algorithm regalg:algorithm1 => Algorithm 1

      Is done. Line 367.

- Line No. 318-326: Fig.10 => Fig.8

Is done. Line 371 to 380

The changes are highlighted in yellow.

  1. Please explain why the mathematical analysis of translation and rotation errors was important to the proposed method and how they were related to the proposed work. Also, why did we need to compare the theoretical translation/rotation tolerance with the experimental tolerances?

To be clearer, we added a descriptive text about your question at the beginning of the section IV, line 220.

The changes are highlighted in green.

  1. Please explain why the results shown in Fig. 4 was important. Is it the common knowledge when a socket is inserted to a fastener? 

Sections 4.2 and 4.3 has been rewriting(lines 234-273), and Fig. 4 and Fig. 5 were modified for a better explanation.

The changes are highlighted in red.

  1. Please include some synthetic images created using the image augmentation library.

Fig. 11 was added. Line 421.

The changes are highlighted in cyan.

Reviewer 2 Report

The manuscript “Robust Fastener Detection Based on Force and Vision Algorithms in Un/Screwing Robotics Applications” describes the use of a collaborative robot that performs diverse screw or unscrew tasks under different environmental changes, i.e., illumination variations. An industrial camera with dynamic focus and active NIR illumination, a mathematical model, a segmentation neural network, and vision to find the center and rotation of the fastener. The document is properly structured, first showing the short introduction related to the problem, followed by the state of art related to robots in screw tasks. Followed by a brief experimental setup. Afterward, diverse results are shown of the segmentation neural network, the algorithm, and the mathematical model, to present the conclusions. The paper shows diverse simulated and real experimental results; therefore, the contributions are relevant to the robotics field.

To improve the manuscript, I suggest to:

1)      This MDPI journal is related to sensors, therefore, please remark on the importance of the sensors used in this paper in the problem statement, i.e., the camera, force, etc.

2)      Section 2 shows the state of the art related to robots in screw tasks, socket wrench, vision systems, and fastener detection. In my opinion, sub-section 3.5 named contributions do not fit in this section, I mean, the authors are describing the contributions of the state of the art presented in sections (2.1-2.4) or the contributions they developed in this research article? These contributions must be placed after the problem statement and the strategies to solve them.

Author Response

Madrid 27/04/2023

Reviewer 2

Sensors (ISSN 1424-8220)

Subject: Response to Reviewer 2 Comments

Thanks for the time to analyze our paper. We must express our gratitude for your insightful comments and suggestions made. The comments have served to improve and complement the research and emphasize the quality of presentation and dissemination of the work.

The comments were responded at the end of this letter, describing the changes we have made and explaining the related problems' relevant aspects.

Additionally, the modified pdf file is added with the changes highlighted in various colors.

  • This MDPI journal is related to sensors, therefore, please remark on the importance of the sensors used in this paper in the problem statement, i.e., the camera, force, etc.

I remarked the use of sensors in the first paragraph of Section 1. Lines 30 to 34. Additionally, the text in the paragraph gives some works using the sensors. Add the table 1 was added.

The changes are highlighted in cyan.

2)      Section 2 shows the state of the art related to robots in screw tasks, socket wrench, vision systems, and fastener detection. In my opinion, sub-section 2.5 named contributions do not fit in this section, I mean, the authors are describing the contributions of the state of the art presented in sections (2.1-2.4) or the contributions they developed in this research article? These contributions must be placed after the problem statement and the strategies to solve them.

 The contributions now are in the Section 3.3. Line 204 with a general description.

The changes are highlighted in red.

Reviewer 3 Report

This paper presents a reasonable method to solve a real application problem. It is well-organized, clearly written, and shows some interesting results that encourage it to be accepted with major revision. However, the commented questions need only to be answered.

1.    Please explicitly indicate and clarify the challenges this study aims to address. What are the challenges and why? Why cannot the previous studies well address these challenges?

2.    At the end of section 1, add a table that summarizes the advantages and disadvantages of existing methods facing the same problem. This way, the reader would rapidly appreciate the novelty of the paper.

3. More studies of detection strategies in un/screwing robotics applications should be cited and discussed.

4.    Fig.7 and Fig.2 need more explanation.

5. Please enrich the captions of all figures and tables for clarification.

6.  Why do you compare with UNET, Mask RCnn resnet50, Deeplab Mobilenet-v2, and Deeplab EdgeMobilenet? There are many state of the arts deep learning structure types, such as YOLO versions, attentions, and the Transformer, that are used with more popular and demonstrate better performance.

7.  In comparison to SOTA methods, more experimental results of other state-of-the-art methods should be given

8.  I also find some grammar problems in this paper. The author needs to carefully check these low mistakes, which is very important for readers.

Author Response

 Madrid 27/04/2023

Reviewer 3

Sensors (ISSN 1424-8220)

Subject: Response to Reviewer 3 Comments

Thanks for the time to analyze our paper. We must express our gratitude for your insightful comments and suggestions made. The comments have served to improve and complement the research and emphasize the quality of presentation and dissemination of the work.

The comments were responded at the end of this letter, describing the changes we have made and explaining the related problems' relevant aspects.

Additionally, the modified pdf file is added with the changes highlighted in various colors. Also, the english editing certificate.

  1. Please explicitly indicate and clarify the challenges this study aims to address. What are the challenges and why? Why cannot the previous studies well address these challenges?

We modified the introduction to explain more clarify the challenges. In the line 30 to 58.

The changes are highlighted in red.

  1. At the end of section 1, add a table that summarizes the advantages and disadvantages of existing methods facing the same problem. This way, the reader would rapidly appreciate the novelty of the paper.

Tab 1 is added. Line 59.

The changes are highlighted in yellow.

  1. More studies of detection strategies in un/screwing robotics applications should be cited and discussed.

We have included three more references [9, 15, 16], please tell us if the cited are enough or could you recommend us some more. Lines (48, 55,57)

The changes are highlighted in gray.

  1. 7 and Fig.2 need more explanation.

More explanation was added into Fig.2 Fig.7

The changes are highlighted in cyan. (lines 233, 362)

  1. Please enrich the captions of all figures and tables for clarification.

Is done.

  1. Why do you compare with UNET, Mask RCnn resnet50, Deeplab Mobilenet-v2, and Deeplab EdgeMobilenet? There are many state of the arts deep learning structure types, such as YOLO versions, attentions, and the Transformer, that are used with more popular and demonstrate better performance.

The goal of this work was developing an industrial robust solution to detect fastener and get the center and vertices. At the beginning the most popular segmentation neural networks (SNN),  UNET and Mask RCNN was performed to know their capabilities, however, this SNN was high time-consumption, therefore, a new searched of SNN carried out, Deeplab allows optimized the time inference and execute directly in some edge devices. It is important in industrial applications, because some industrial PC are not powerful and some of them does not allow insert a GPU. This work presents a comparative of the SNN to gives a reference for futures applications where the time-inference and hardware resources are important. The results are presented in the Table 3 and Table 4.

The tables are highlighted in green. (line 333)

Some Yolo versions, attentions, transformer, UNET and Mask RCNN have some layers with specific operations that cannot applied a direct optimization to increase the time inference, a rewrite or create new layers are required, and it is out of this work.

  1. In comparison to SOTA methods, more experimental results of other state-of-the-art methods should be given

At the time point, Deeplab EdgeMobilenet has been demonstrated a very well performance in the applications developed. Additionally, for us nowadays is impossible carried out more experiments. However, in the future we could be realized a study and provide a publication about the review and extensive comparison of SOTA segmentation neural networks.

  1. I also find some grammar problems in this paper. The author needs to carefully check these low mistakes, which is very important for readers.

English native speaker has done an extensive linguistic revision of the article, correcting several syntax and grammar errors in the manuscript. The certificate is attached.

Reviewer 4 Report

This paper is in line with the scope of this journal with some significant contributions, but I am concerned about the following questions:

l  The authors should further analyze the results, In conclusion, I think the author's analysis of the research results is very shallow, so they must further summarize the research findings based on the research results

l   Please define all acronyms in the abstract. - State clearly the research question in the Introduction. The authors need to add also info about the result of the research.

l   I think that not all symbols are defined for equations, especially in the part where a novel method is presented.

lExpend conclusion to include details regarding future work.

l   There are some technical and English language errors, please read the manuscript carefully and revise.

l  Please remove “we” from the manuscript and instead use the proposed system.

Author Response

         Madrid 27/04/2023

Reviewer 4

Sensors (ISSN 1424-8220)

Subject: Response to Reviewer 4 Comments

I hope you are fine and auguring success in your research and work activities.

Thanks for the time to analyze our paper. I must express my gratitude for your insightful comments and suggestions made. The comments have served to improve and complement the research and emphasize the quality of presentation and dissemination of the work.

The comments were responded at the end of this letter, describing the changes we have made and explaining the related problems' relevant aspects.

Additionally, the modified pdf file is added with the changes highlighted in various colors. Also, the English editing certificate.

  1. The authors should further analyze the results, In conclusion, I think the author's analysis of the research results is very shallow, so they must further summarize the research findings based on the research results

The conclusions were rewriter to be clearer. (line 485).

The changes are highlighted in yellow.

  1. Please define all acronyms in the abstract. - State clearly the research question in the Introduction. The authors need to add also info about the result of the research.

Acronyms were written in the abstract and the section abbreviations in line 578.

The changes are highlighted in cyan.

  1. I think that not all symbols are defined for equations, especially in the part where a novel method is presented.

Symbols are defined in section 6 and Figure 7.

The changes are highlighted in red.

  1. Expend conclusion to include details regarding future work.

Just done. (line 551) The changes are highlighted in green.

  1. There are some technical and English language errors, please read the manuscript carefully and revise.

English native speaker has done an extensive linguistic revision of the article, correcting several syntax and grammar errors in the manuscript. The Certifícate is attached.

  1. Please remove “we” from the manuscript and instead use the proposed system.

Just done.

Reviewer 5 Report

Dear Authors,

Please find the attached file for review comments.

Author Response

Madrid 27/04/2023

Reviewer 5

Sensors (ISSN 1424-8220)

Subject: Response to Reviewer 5 Comments

I hope you are fine and auguring success in your research and work activities.

Thanks for the time to analyze our paper. I must express my gratitude for your insightful comments and suggestions made. The comments have served to improve and complement the research and emphasize the quality of presentation and dissemination of the work.

The comments were responded at the end of this letter, describing the changes we have made and explaining the related problems' relevant aspects.

Additionally, the modified pdf file is added with the changes highlighted in various colors. Also, the English editing certificate.

1 Formulation missing for IMU data interpretation in the proposed system. How and where does IMU data were utilized ?

IMU are used to read random values rx with respect to the robot axis when real tests are performed. The description was mentioned in line 183 and figure 1.

The changes are highlighted in green.

  1. As it indicated in Figure 11, is GPS solution used for the system?

Fig 12 (line 435) was updated. GPS is not used; the embedded device has a GPS and Imu integrated.

The changes are highlighted in cyan.

  1. How does proposed approach overcome vibration caused by the wrench dynamics while on the operation.

In the proposed solution an iterative algorithm searches for the center and rotation of the screw, the offset produced is corrected by the robot's movement until the center of the camera and the screw match. Therefore, the dynamics of the socket wrench does not affect the search and insertion.

  1. In the table figures, there are cells left empty in Table 5 and 6, but couldn't understand actual meaning of the empty cells. Instead authors could use some notation and give small description at the bottom of table.

Just done. In the upload document Tables 7 and 8 are updated. (line 485).

The changes are highlighted in red.

5 overall major contributions of proposed approach is to vision based fastener detection, but there were no any comparison provided to evaluate the significance of proposed system against existing system. Authors can find some related works in literature as there are some similar solutions already exists. And it is very meaningful to discuss, in what aspects proposed system is more efficient?

In the literature does not exist a normalized method or standard testbed to compare the performance algorithms is the screwing tasks. Other works propose methodologies and algorithms according to the specific tasks requirements.

The algorithms and methodology proposed has been tested in simulation and a real task, obtained an accuracy of 96% finding the center and rotation of the fastener in real experiment. We don’t have any equivalent performance index in the literature. In or work the results are presented in the Table 7, 8, 9.

Round 2

Reviewer 3 Report

The manuscript is well-revised, and it is acceptable in its current form.